# Impaired Humoral Response in Renal Transplant Recipients to SARS-CoV-2 Vaccination with BNT162b2 (Pfizer-BioNTech)

**DOI:** 10.3390/v13050756

**Published:** 2021-04-25

**Authors:** Johannes Korth, Michael Jahn, Oliver Dorsch, Olympia Evdoxia Anastasiou, Burkhard Sorge-Hädicke, Ute Eisenberger, Anja Gäckler, Ulf Dittmer, Oliver Witzke, Benjamin Wilde, Sebastian Dolff, Andreas Kribben

**Affiliations:** 1Department of Nephrology, University Hospital Essen, University of Duisburg-Essen, Hufelandstr. 55, 45147 Essen, Germany; michael.jahn@uk-essen.de (M.J.); ute.eisenberger@uk-essen.de (U.E.); anja.gackler@uk-essen.de (A.G.); Benjamin.wilde@uk-essen.de (B.W.); andreas.kribben@uk-essen.de (A.K.); 2KfH Kuratorium für Dialyse und Nierentransplantation e.V, KfH-Nierenzentrum Friesener Straße 37a, 96317 Kronach, Germany; oliver.dorsch@kfh.de; 3Institute for Virology, University Hospital Essen, University of Duisburg-Essen, Virchowstr. 179, 45147 Essen, Germany; olympiaevdoxia.anastasiou@uk-essen.de (O.E.A.); ulf.dittmer@uk-essen.de (U.D.); 4KfH Kuratorium für Dialyse und Nierentransplantation e.V, KfH-Nierenzentrum Alfried-Krupp-Str. 43, 45131 Essen, Germany; burkhard.sorge-haedicke@kfh-dialyse.de; 5Department of Infectious Diseases, West German Centre of Infectious Diseases, University Hospital Essen, University of Duisburg-Essen, Hufelandstr. 55, 45147 Essen, Germany; oliver.witzke@uk-essen.de (O.W.); sebastian.dolff@uk-essen.de (S.D.)

**Keywords:** SARS-Cov-2 vaccination, renal transplant recipients, renal transplantation, COVID-19

## Abstract

The severe acute respiratory syndrome coronavirus-2 (SARS-CoV-2) has a major impact on transplant recipients, with mortality rates up to 20%. Therefore, the effect of established messenger RNA (mRNA)-based SARS-CoV-2 vaccines have to be evaluated for solid organ transplant patients (SOT) since they are known to have poor responses after vaccination. We investigated the SARS-CoV-2 immune response via SARS-CoV-2 IgG detection in 23 renal transplant recipients after two doses of the mRNA-based SARS-CoV-2 vaccine BNT162b2 following the standard protocol. The antibody response was evaluated once with an anti-SARS-CoV-2 IgG CLIA 15.8 +/− 3.0 days after the second dose. As a control, SARS-CoV-2 IgG was determined in 23 healthcare workers (HCW) and compared to the patient cohort. Only 5 of 23 (22%) renal transplant recipients were tested positive for SARS-CoV-2 IgG antibodies after the second dose of vaccine. In contrast, all 23 (100%) HCWs were tested positive for antibodies after the second dose. Thus, the humoral response of renal transplant recipients after two doses of the mRNA-based vaccine BNT162b2 (Pfizer-BioNTech, Kronach, Germany) is impaired and significantly lower compared to healthy controls (22% vs. 100%; *p* = 0.0001). Individual vaccination strategies might be beneficial in these vulnerable patients.

## 1. Introduction

The novel severe acute respiratory syndrome coronavirus-2 (SARS-CoV-2) is a major threat to solid organ transplant recipients (SOT) with a mortality rate of up to 20% [1,2]. Currently there is a lack in efficient treatment options [3]. New messenger RNA (mRNA)-based SARS-CoV-2 vaccines were evaluated in over 70,000 individuals and found to have an efficacy of 95% in phase 3 placebo-controlled trials [4,5]. Since December 2020, the European union approved the first mRNA-based SARS-CoV-2 vaccine of Pfizer-BioNTech (https://www.pei.de/DE/home/home-node.html, accessed on 21 December 2020). Immunocompromised patients, like SOT recipients were not included in the large phase 3 trials, and therefore, efficacy data are lacking. However, vaccination is recommended for SOT patients in Germany [6,7]. The current study evaluates the immune response of 23 renal transplant recipients after standard vaccination protocol with two vaccinations with the mRNA-based SARS-CoV-2 vaccine BNT162b2 (Pfizer-BioNTech, Nierenzentrum Kronach) in January and February 2021 by SARS-CoV-2 IgG development. The results were compared to the antibody response of 23 healthcare workers vaccinated using the same standard protocol with the same vaccine [8].

## 2. Materials and Methods

Twenty-three renal transplant recipients of the Nierenzentrum Kronach, Germany were intramuscularly vaccinated twice with a gap of 22.0 +/− 4.6 days between the shots with the mRNA-based SARS-CoV-2 vaccine BNT162b2 (Pfizer-BioNTech) according to the standard protocol at a vaccination center in Kronach, Germany [9]. Fourteen days after the second vaccination serum samples were tested for SARS-CoV-2 IgG against the Spike glycoprotein using an approved anti-SARS-CoV-2 IgG CLIA (LIAISON^®^ SARS-CoV-2 TrimericS IgG assay, Diasorin, Saluggia, Italy). According to the manufacturer’s recommendations for the Chemiluminescence Enzyme Immunoassays (CLIA), an Arbitrary Units per milliliter (AU/mL) ratio of <13.0 was considered to be negative and ≥13.0 to be positive. A conversion of AU/mL to binding antibody units (BAU/mL) that correlate with the WHO standard is possible using the following equation: BAU/mL = 2.6*AU/mL. 800.0 AU/mL (2080 BAU/mL) is the upper limit of quantification without dilution of the CLIA.

In addition, the antibody response was compared to 23 healthcare workers after two intramuscular vaccinations at the University Hospital Essen in January 2021 with the same vaccination, sampling and testing protocol as the renal transplant recipients. The HCW received regular testing with teal-time PCR-assays for SARS-CoV-2 RNA from nasal swabs and had no clinical suspicion for SARS-CoV-2 infections throughout the preceding 12 months. Since the current study was focused on the humoral immune response in renal transplant patients, T-cell activity was not evaluated in this study. Fisher’s exact test and Mann–Whitney U test was used to compare the results between groups.

## 3. Results

Of the 23 renal transplant recipients included in the study, none had a prior or current diagnosis of COVID-19. The mean age was 57.7 +/− 13.5 years. Twelve (52%) of the 23 patients were female and 11 (48%) were male. The mean time after renal transplantation was 11.4 +/− 9.2 years. The immunosuppressive regimen included mycophenolate (18 of 23; 78%), tacrolimus (14 of 23; 60%) and corticosteroids (14 of 23; 60%) (Table 1). None of the 23 HCW had a confirmed diagnosis of COVID-19 prior to the vaccination. The mean age was 44.4 +/− 9.2 years. Fourteen (61%) were female and 9 (39%) were male. Five of 23 (22%) renal transplant recipients tested positive for SARS-CoV-2 IgG at a mean of 15.8 +/− 3.0 days after the second dose of vaccine (Table 1). The mean SARS-CoV-2 IgG titer was 50.9 +/− 138.7 AU/mL. 

All 23 (100%) HCW tested positive for SARS-CoV-2 IgG at a mean of 13.7 +/− 1.8 days after the second dose. The mean SARS-CoV-2 IgG titer was 727.7 +/− 151.3 AU/mL.

The immunosuppressive regimens in patients who tested positive and negative for SARS-CoV-2 IgG antibodies after vaccination were similar and there were no differences in age, gender and immunosuppressive drugs between these groups of patients (Table 2).

The proportion of renal transplant recipients testing positive for antibodies after protocol-based vaccination was significantly lower in comparison to the control group of HCW (22% vs. 100%, *p* = 0.0001, Figure 1). In addition, the mean SARS-CoV-2 antibody titer of the renal transplant recipients was significantly lower in comparison to the HCW (50.9 +/− 138.7 AU/mL vs. 727.7 +/− 151.3 AU/mL, *p* = 0.0001). 

## 4. Discussion

This is the first study which evaluates the immune response of 23 renal transplant recipients after standard protocol-based vaccination of two doses of the mRNA-based SARS-CoV-2 vaccine BNT162b2. The immune response was evaluated by SARS-CoV-2 IgG anti-Trimeric Spike glycoprotein detection 15.8 +/− 3.0 days after the second dose of the vaccine. The results were compared to a control group of 23 healthy healthcare workers after using the same standard protocol with two doses of the same mRNA-based SARS-CoV-2 vaccine BNT162b2. Only 5 of the 23 (22%) renal transplant recipients tested positive for SARS-CoV-2 IgG antibodies after vaccination. In comparison all the 23 HCW (100%) tested positive for SARS-CoV-2 IgG antibodies after vaccination (22% vs. 100%, *p* = 0.0001). In addition, the mean SARS-CoV-2 IgG titer of renal transplant recipients was significantly lower (50.9 +/− 138.7 vs. 727.7 +/− 151.3, *p* = 0.0001). The data exposes the impaired immune response of solid organ transplant (SOT) patients after standard mRNA-based SARS-CoV-2 vaccination. It is likely that SOT patients need an individualized vaccination scheme. This may include more than two booster doses or even a combined scheme with mRNA vaccines, protein/subunit vaccines and vector-based vaccines. The findings are in line with the impaired responses to other vaccination in patients after SOT and is most likely due to immunosuppression [10]. Our results are concordant with those from a recently published study by Boyarsky et al., which evaluated immunogenicity in 436 SOT after one dose of an mRNA-based vaccine. The authors observed that only 76 of 436 of the SOT patients (17%) had a positive antibody response [11]. In addition a recently published study by Chavarot and colleagues evaluated the IFNγ T-cell responses after two injections of an mRNA vaccine in kidney transplant recipients treated with belatacept [12]. The authors reported a low seroconversion rate at day 60 and a T-cell response in only 30.4% of the patients measured. The current data suggest that these patients might be vulnerable for COVID-19 disease in spite of their vaccination status, since reliable immunization after protocol-based vaccination with the mRNA-based SARS-CoV-2 vaccine BNT162b2 was not verifiable. It might be possible that other approved vaccines such as vector-based vaccines induce better humoral responses, but further studies are needed to address this issue. A limitation of the present study is the lack of results regarding SARS-CoV-2 specific T-cell response and neutralization capacity of the sera, which could counterbalance the impaired humoral response. In clinical practice, booster doses or higher initial dosages might improve the immune response in SOT patients such as in diphtheria, hepatitis B or pneumococcal vaccination [13,14].

## 5. Conclusions

In conclusion, the detectable humoral immune response after standard protocol vaccination with two doses of the mRNA-based SARS-CoV-2 vaccine BNT162b2 in 22% of all renal transplant patients is poor. We suggest that renal transplant recipients should be monitored for immune responses, and novel individual vaccination strategies might be needed and evaluated in clinical trials in this vulnerable cohort. Nevertheless, studies with more participants and various vaccine candidates are needed to evaluate the different effects and further immune responses like B-and T-cell responses in patients after SOT.

## Figures and Tables

**Figure 1 viruses-13-00756-f001:**
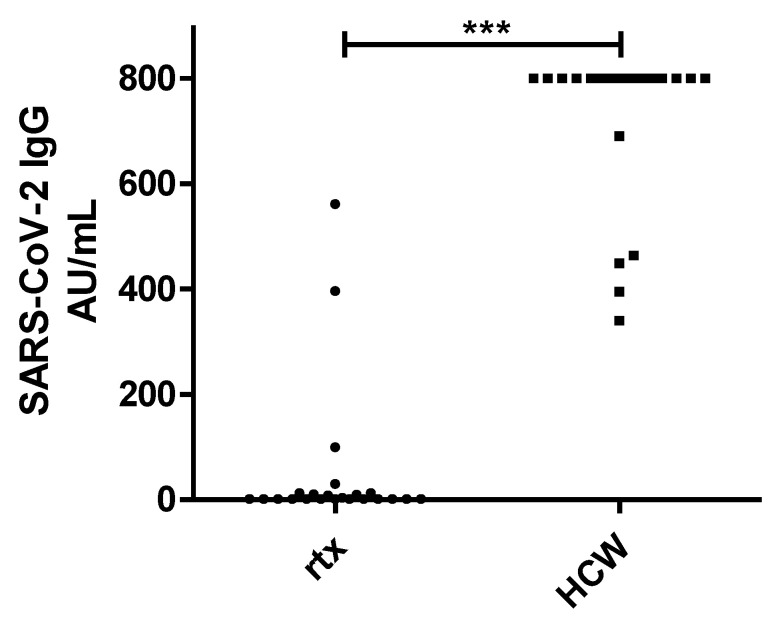
IgG Chemiluminescence Enzyme Immunoassays (CLIA) Arbitrary Units per milliliter (AU/mL) ratio in 23 renal transplant recipients (rtx) and 23 healthcare workers (HCW) after two doses of the mRNA-based SARS-CoV-2 vaccine BNT162b2. *** *p* < 0.0001.

**Table 1 viruses-13-00756-t001:** Characteristics of patients after renal transplantation and healthcare workers after two doses of the mRNA-based SARS-CoV-2 vaccine BNT162b2. rtx renal transplant recipients; HCW, healthcare workers; n number; pos positive; neg negative; Ab antibody, CLIA Chemiluminescence Enzyme Immunoassays; AU Arbitrary Units; mL milliliter; Ab antibody.

	rtx	HCW	
**n**	23	23	*p*
**female/male (n; %)**	12 (52%)/11 (48%)	14 (61%)/9 (39%)	0.76
**age (years)**	57.7 +/− 13.5	44.4 +/− 9.2	0.0003
**immunosuppression (n)**		--	
**mycophenolate n (%)**	18, (78%)		
**corticosteroids n (%)**	14 (60%)		
**tacrolimus n (%)**	14 (60%)		
**cyclosporine n (%)**	4 (17%)		
**sirolimus n (%)**	5 (22%)		
**everolimus n (%)**	1 (4%)		
**belatacept n (%)**	1 (4%)		
**azathioprine n (%)**	1 (4%)		
**years after rtx**	11.4 +/− 9.2	--	
**days between first and second dose (days)**	22.0 +/− 4.6	22.0 +/− 0	
**SARS-CoV-2 Ab detection after second dose (days)**	15.8 +/− 3.0	13.7 +/− 1.8	
**SARS-CoV-2 Ab pos** **CLIA (n; %)**	5 (22%)	23 (100%)	0.0001
**SARS-CoV-2 Ab neg** **CLIA (n; %)**	18 (78%)	0 (0%)	
**Ab SARS-COV-2 CLIA (AU/mL)**	50.9 +/− 138.7	727.7 +/− 151.3	0.0001

**Table 2 viruses-13-00756-t002:** Characteristics of renal transplant recipients who tested positive and negative for SARS.CoV-2 IgG after two doses of the mRNA-based SARS-CoV-2 vaccine BNT162b2.

	SARS-CoV-2 IgG Positive	SARS-CoV-2 IgG Negative
**n**	5	18
**female/male n (%)**	3 (60%)/2 (40%)	9 (50%)/9 (50%)
**age (years)**	57.0 +/− 8.1	57.9 +/− 14.9
**time after rtx (years)**	17.6 +/− 7.7	9.7 +/− 9.1
**mycophenolate n (%)**	3 (60%)	15 (83%)
**corticosteroids n (%)**	3 (60%)	11 (61%)
**tacrolimus n (%)**	2 (40%)	12 (67%)
**cyclosporine n (%)**	2 (40%)	2 (11%)
**sirolimus n (%)**	1 (20%)	4 (22%)
**everolimus n (%)**	1 (20%)	0
**betalacept n (%)**	0	1 (6%)
**azathioprine n (%)**	0	1 (6%)
**number of immunosuppressive drugs n (%)**	2.4 +/− 0.5	2.6 +/− 0.5

## Data Availability

The data that support the findings of this study are available from the corresponding author upon reasonable request.

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
