# Peer review of "Impaired Humoral Response in Renal Transplant Recipients to SARS-CoV-2 Vaccination with BNT162b2 (Pfizer-BioNTech)"

_viruses, 2021, doi:10.3390/v13050756_

Round 1

Reviewer 1 Report

In this manuscript, Korth et al. investigated the SARS-CoV-2 humoral immune response in SOT patients. The presented data is convincing and clear. I just have some small questions/comments.

  • Please check the manuscript for spelling and grammar mistakes
  • Lines 80-81: None of the 23 HCW had a confirmed diagnosis of COVID-19 prior to the vaccination
    à Were they ever tested or were they maybe asymptomatic; please clarify in text
  • Is it possible to look at neutralizing capacity of the serum? Is the amount of SARS-CoV2 antibodies raised in the SOT patients enough the neutralize the virus?
  • Lines 139-140: A limitation of this present study is the lack of results regarding SARS-CoV- 2 specific T-cell response
    à Can the author explain why they did not study this? Is anything known about this in literature?
  • All the SOT patients were vaccinated with an mRNA based vaccine, this is also the case for the Boyarsky study (Ref 11). What about the vaccines with adenoviral vectors like Astrazeneca? Do the authors expect the same type of response or would this be different?
  • Did you look at the levels in between doses, so after the first dose like in the Boyarsky study? Would you expect the levels to increase in SOT patients after the second dose?

Author Response

We thank Reviewer 1 for the detailed review of the manuscript. Based on your comments we revised the manuscript.

Please find attached the point-by-point reply.

Comments and Suggestions for Authors

In this manuscript, Korth et al. investigated the SARS-CoV-2 humoral immune response in SOT patients. The presented data is convincing and clear. I just have some small questions/comments.

Please check the manuscript for spelling and grammar mistakes

We corrected spelling and grammar mistakes.

Lines 80-81: None of the 23 HCW had a confirmed diagnosis of COVID-19 prior to the vaccination
à Were they ever tested or were they maybe asymptomatic; please clarify in text 

We clarified the selection of healthcare workers for the first SARS-CoV-2 vaccinations performed at the university hospital Essen in the introduction section:

“The HCW received regular testing with teal-time PCR-assays for SARS-CoV-2 RNA from nasal swaps and had no clinical suspicion for SARS-CoV-2 infections throughout the preceding 12 months.”

Is it possible to look at neutralizing capacity of the serum? Is the amount of SARS-CoV2 antibodies raised in the SOT patients enough the neutralize the virus

We added the lack of neutralization capacity to the limitations:

“A limitation of the present study is the lack of results regarding SARS-CoV-2 specific T-cell response and neutralization capacity of the sera, which could counterbalance the impaired humoral response.”

Lines 139-140: A limitation of this present study is the lack of results regarding SARS-CoV- 2 specific T-cell response
 Can the author explain why they did not study this?

Thank you for your suggestion. We added the following statement addressing the initial focus of the study in the methods section.

“Since the current study was focused on the humoral immune response in renal transplant patients, T-cell activity was not evaluated in this study”.

Is anything known about this in literature?

We added the recently published literature to the discussion section:

In addition a recently published study by Chavarot and colleagues evaluated the IFNγ T-cell responses after two injections of mRNA vaccine in kidney transplant recipients treated with belatacept [1]. The authors reported a low seroconversion rate at days 60 and a T-cell response in only 30.4% of the patients measured.”

All the SOT patients were vaccinated with an mRNA based vaccine, this is also the case for the Boyarsky study (Ref 11). What about the vaccines with adenoviral vectors like Astrazeneca? Do the authors expect the same type of response or would this be different?

The authors expectation was added to the discussion section:

“It might be possible that other approved vaccines such as vector-based vaccines induce better humoral responses but further studies are needed to address this issue.”

Did you look at the levels in between doses, so after the first dose like in the Boyarsky study? 

The study design contained the evaluation after the second dose of vaccine, since two doses are needed for efficient vaccination:

Fourteen days after the second vaccination serum samples were tested for SARS-CoV-2 IgG against the Spike glycoprotein using an approved anti-SARS-CoV-2 IgG CLIA (LIAISON® SARS-CoV-2 TrimericS IgG assay, Diasorin, Saluggia, Italy).“

Would you expect the levels to increase in SOT patients after the second dose? 

We added the expectation to the discussion section:

“It is likely that SOT patients need an individualized vaccination scheme. This may include more than two booster doses or even a combined scheme with mRNA vaccines, protein/subunit vaccines and vector-based vaccines.”

Reviewer 2 Report

The current article by Johannes Korth and al., reviews the current literature on the correlation between SARS-CoV-2 vaccination with BNT162b2 (Pfizer-BioNTech) and impaired humoral response in renal transplant recipients.The title of the paper is in line with the body of the manuscript.The topic is current and timely and very important for the world's scientific community in the current period of the global COVID-19 pandemic. The authors have written a clear and detailed review and the material is well presented, although the authors do not express particular personal ideas on the future prospects for investigation. The references used are suitable and it is new and updated material, I believe that the article can be accept pending other revision, even if I have below some observations that I think they can make the paper more complete:

  1. The authors should provide a more personal opinion of the results obtained

  2. The authors should include in their conclusions a personal opinion if varying the dose these patients they cuold have an antibody response

  3. The authors should clear if think this answer is the same for all types of vaccines or just mRNA vaccines

  4. In the conclusions the authors should include suggestions on how to set up new clinical trials

Author Response

We thank Reviewer 2 for the detailed review of the manuscript. Based on your comments we revised the manuscript.

Please find attached the point-by-point reply.

Comments and Suggestions for Authors

The current article by Johannes Korth and al., reviews the current literature on the correlation between SARS-CoV-2 vaccination with BNT162b2 (Pfizer-BioNTech) and impaired humoral response in renal transplant recipients.The title of the paper is in line with the body of the manuscript.The topic is current and timely and very important for the world's scientific community in the current period of the global COVID-19 pandemic. The authors have written a clear and detailed review and the material is well presented, although the authors do not express particular personal ideas on the future prospects for investigation. The references used are suitable and it is new and updated material, I believe that the article can be accept pending other revision, even if I have below some observations that I think they can make the paper more complete:

  1. The authors should provide a more personal opinion of the results obtained 

We added the authors implications of the results to the conclusion:

We suggest that renal transplant recipients should be monitored for immune responses, and novel individual vaccination strategies might be needed and evaluated in clinical trials in this vulnerable cohort.”

  1. The authors should include in their conclusions a personal opinion if varying the dose these patients they could have an antibody response

We included the authors opinion in the discussion section.

“In clinical practice, booster doses or higher initial dosages might improve the immune response in SOT patients such as in diphtheria, hepatitis B or pneumococcal vaccination.”

In addition, the conclusion section was extended as mentioned before.

We suggest that renal transplant recipients should be monitored for immune responses, and novel individual vaccination strategies might be needed and evaluated in clinical trials in this vulnerable cohort.”

  1. The authors should clear if think this answer is the same for all types of vaccines or just mRNA vaccines 

We added the authors opinion to the discussion section:

It might be possible that other approved vaccines such as vector-based vaccines induce better humoral responses, but further studies are needed to address this issue. “

  1. In the conclusions the authors should include suggestions on how to set up new clinical trials.

The authors included suggestions for new clinical trials in the limitations section:

A limitation of the present study is the lack of results regarding SARS-CoV-2 specific T-cell response and neutralization capacity of the sera, which could counterbalance the impaired humoral response.”

In addition, we extended the conclusions with the authors expectations for clinical evaluations:

“We suggest that renal transplant recipients should be monitored for immune responses, and novel individual vaccination strategies might be needed and evaluated in clinical trials in this vulnerable cohort.”

References:

  1. Chavarot, N.; Ouedrani, A.; Marion, O.; Leruez-Ville, M.; Villain, E.; Baaziz, M.; Del Bello, A.; Burger, C.; Sberro-Soussan, R.; Martinez, F.; et al. Poor Anti-SARS-CoV-2 Humoral and T-cell Responses After 2 Injections of mRNA Vaccine in Kidney Transplant Recipients Treated with Belatacept. Transplantation 2021, Publish Ah, doi:10.1097/TP.0000000000003784.